# Silver Nanoparticles and Its Mechanistic Insight for Chronic Wound Healing: Review on Recent Progress

**DOI:** 10.3390/molecules27175587

**Published:** 2022-08-30

**Authors:** Manoj Singh, Vanita Thakur, Vikas Kumar, Mayank Raj, Shivani Gupta, Nisha Devi, Sushil Kumar Upadhyay, Markéta Macho, Avik Banerjee, Daniela Ewe, Kumar Saurav

**Affiliations:** 1Department of Biotechnology, Maharishi Markandeshwar (Deemed to be University), Mullana 133207, India; 2Laboratory of Algal Biotechnology-Centre Algatech, Institute of Microbiology of the Czech Academy of Sciences, 37901 Třeboň, Czech Republic

**Keywords:** wound, wound healing, silver nanoparticles, dermatology, inflammatory response

## Abstract

Wounds are structural and functional disruptions of skin that occur because of trauma, surgery, acute illness, or chronic disease conditions. Chronic wounds are caused by a breakdown in the finely coordinated cascade of events that occurs during healing. Wound healing is a long process that split into at least three continuous and overlapping processes: an inflammatory response, a proliferative phase, and finally the tissue remodeling. Therefore, these processes are extensively studied to develop novel therapeutics in order to achieve maximum recovery with minimum scarring. Several growth hormones and cytokines secreted at the site of lesions tightly regulates the healing processes. The traditional approach for wound management has been represented by topical treatments. Metal nanoparticles (e.g., silver, gold and zinc) are increasingly being employed in dermatology due to their favorable effects on healing, as well as in treating and preventing secondary bacterial infections. In the current review, a brief introduction on traditional would healing approach is provided, followed by focus on the potential of wound dressing therapeutic techniques functionalized with Ag-NPs.

## 1. Introduction

A wound is described as any disruption in the anatomic structures, function, or any break in the continuity of the skin [1,2]. Cuts, scares, and scratches are common, and the skin loses morphological traits and functions in the damaged region. Wounds are sometimes called as “silent epidemics” because if they are not treated, they can result in the loss of limbs, legs, or even death [3]. About 8.2 million medicare patients suffer from chronic, nonhealing wounds. Medicare’s cost estimates for all wounds varied from $28.1 billion to $96.8 billion, including infection control expenditures, with surgical and diabetic ulcers being the most expensive to treat [4]. Outpatient expenditures ($9.9–$35.8 billion) were also greater than inpatient costs ($5.0–$24.3 billion), probably due to an increase in current outpatient treatments [4]. Despite these alarming figures, research funding is inadequate in comparison to the threat of these severe health problems [5,6]. Diabetic foot ulcers, for example, have a death rate of 30.5% that is equivalent to worldwide death rate of cancer (31%), which has remained stable since 2007, even though funding for research into these life-threatening consequences is substantially lower than investment for cancer research [7]. Venous ulcers (VUs) are ulcers caused by chronic venous insufficiency that mostly affect the lower limbs [8]. VU research at National Health Service (NHS), United Kingdom estimated that one in every 500 people has VU, costing NHS around £400–600 million towards health care expenses. An estimated 500,000–600,000 persons in the United States suffers with venous leg ulcers, costing the health-care system over $1 billion annually [9].

Many scientists have predicted that a wound takes around 6–8 weeks to heal. The presence of a foreign pathogen, prolonged irritation, trauma, and ischemia are some of the causes of chronic wound healing [10,11]. Chronic injuries do not recover gradually. Hypoxia, pH shift, and bacterial colonization are only a few of the pathophysiologic variables that cause prolonged healing. Chronic wounds (Figure 1) affect a large portion of the population and has a detrimental influence on individual’s quality of life.

## 2. Wound Infection Development and Its Progression

All skin injuries or illnesses induced by trauma or therapeutic treatments are referred to as wounds. Wounds can be divided into two categories, open and closed, which are created by cutting, hitting, or burning [12]. Open types are commonly associated with bleeding and rupturing of skin layers [13,14,15,16], whereas closed wounds are produced by bruises, dead blood, or impacts. Based on their clinical features, a wound can be acute or chronic. Burns, cuts, and surgical incisions generate acute wounds, which need 8–12 weeks to heal [14], while chronic wounds can last for much longer time and are triggered by a number of causes, e.g., bacterial infection. Bacteria are naturally present in the skin and wounds; 10^5^ bacteria have been identified as the essential threshold between colonization and clinically significant illness [17]. They can enter the underlying tissues of injured skin, triggering inflammation and the production of proteases and reactive oxygen species by inflammatory cells [18,19]. Bacterial endotoxins raise cytokine levels while lowering growth factor production and collagen synthesis [20]. The infection can then advance from contamination to colonization, which is inhibited by biofilm formation [21]. Biofilms were found in 60% of chronic and 6% of acute wound biopsy samples [18]. In infected wounds, Bessa et al. [22] detected bacterial infections as well as treatment resistance profiles. The researchers gathered 312 wound swab samples from 213 patients suffering from various sorts of infections, out of which 28 isolates were identified and characterized such as *S. aureus*, *P. aeruginosa*, *P. mirabilis*, *E. coli* and *Corynebacterium* spp. [22]. *P. aeruginosa* possesses a variety of resistance mechanisms that reduce permeability, synthesizes antibiotic-inactivating enzymes, and change target proteins [23]. The most common cause of drug resistance is due to the excessive use and misuse of incompatible antibiotics [24], Table 1.

## 3. Traditional Wound Healing Approach

Wound healing is a complicated and carefully controlled process which involves the regeneration, reformation, and repair of injured tissues [49], all of which are highly necessary for the skin’s barrier function to be retained. Homeostasis, inflammation, proliferation, and remodeling are the four phases required for healing. Platelets, which are also known as inflammatory cells, are the first cells to arrive at a lesion site. They are activated by the breakdown of collagens and the production of thrombin, which initiates the coagulation cascade, resulting in the formation of a fibrin clot at the damage site. Platelets also activate the complement system, which leads to the synthesis of platelet-derived growth factors (PDGF), transforming growth factors (TGF), interleukin, and insulin growth factors. Platelets also form a cluster in the event of an injury, which prevents blood flow. By interacting with fibroblasts, connective tissue cells that deposit collagen and prevent bleeding and platelet-derived growth factors (PDGF) increase vascularization [50]. 

The inflammatory phase lasts six days and begins immediately after the damage is caused [51]. Endothelial cells adhere to the injury and release cell adhesion molecules (CAMs) which function as hooks, and more neutrophils attach onto the endothelial cell surface and push against permeable cell connections via mast cell mediators [52,53] and triggering the inflammatory cell response. After 24–48 h, the site of lesions produces numerous vasoactive mediators and chemotactic factors and mast cells release enzymes, histamine, and other active amines, resulting in an inflammatory response. Neutrophils are the first cells to arrive at the injury site, attaching to endothelial cells within 24 h and becoming active. Pathogens, foreign substances, and dead cells are all phagocytized by neutrophils. The inflammatory response alerts neutrophils, which start attaching to the endothelial cells at the site of the injury within 24 h of infection and phagocytizing pathogens, foreign substances, and dead cells. In roughly two days, monocytes and lymphocytes migrate to the site of the injury and transform into macrophages, removing necrotic tissue and pathogens and starting the formation of granulation tissue [54]. Macrophages produce a protease inhibitor, which helps them to release neutrophils [55] as well as growth factors including TGF, PDGF, tumor necrosis factor (TNF), and cytokines, which are required for fibroblast, smooth muscle cell, endothelial cell proliferation and extracellular macromolecules (ECM) and minerals, deposition [56]. 

The proliferative phase begins with the repair of injured tissues and the initiation of angiogenesis through ECM synthesis. This stage lasts for around 2–3 days following the injury and continues until the wound heals. Fibroblasts and endothelial cells are the primary cells at this stage. Angiogenesis, which is required to produce granulation tissue, is stimulated by vascular endothelial growth factors A (VEGF-A), FGF-2, PDGF, and TGF. Inflammatory cytokines like interleukin-1 (IL-1) and TNF-a cause fibroblasts to release growth factors such as EGF, KGF, and HGF, which attract keratinocytes to the wound bed and cause granulation tissue to develop [57]. From wound borders and skin appendages, basal keratinocytes move to the injured area, proliferate, differentiate, and eventually form a covering around it. Contraction occurs when the edges of the area are pulled together by fibroblasts in the wound bed to seal the surface [58]. 

The final stage is maturation and reconstruction, which begin three days after the damage and can extend up to two years. Endothelial cells, myofibroblasts, and macrophages are all killed or removed from the surface [58]. Small capillaries combine to generate bigger blood vessels and metabolic wound-healing activity decreases. The bulk of the ECM in a damaged area is made up of collagen and other ECM proteins. Fibroblasts generate lysyl oxidases which realigns collagen into an organized network, increasing tensile strength to about 80% of healthy tissue. Matrix metalloproteinase (MMPs), which are secreted by fibroblasts and other cells, control the collagen formation [59,60] (Figure 2).

Chronic wounds harbor variety of microorganisms due to their synergistic bacterial growth [61], which can sometimes perform a positive role in limiting the growth of other germs. Once the microbes reach the infection site, they disrupt the healing process and create biofilms. Methicillin-resistant *Staphylococcus aureus* (MRSA) and *S. aureus* are the most prevalent bacteria that impair the wound healing at early stage, while *E. coli* and *P. aeruginosa* infect the deeper layers of the skin. The resources are consequently being shared among the microbes that support each other’s proliferation, due to which *S. aureus* and *P. aeruginosa* grows on multiple wounds at the same time [19]. Given the variety of different bacterial species that survive in the wounds side by side, there is an urgent need for new antimicrobial agents to address the global concern of increase in antimicrobial resistance (AMR) [62]. New approach to overcome AMR are being continuously researched, including various medicinal plant extracts [63], combination of effective antibiotics and other antibacterial materials [64,65], especially nano based materials, which comprise unary [66], binary (Ag/Au, Ag/Pt nanoparticles) [66,67] or multi-component meso-crystalline materials (CdZnSe-CdZnS nanoalloy) [68]. 

Since 1970s, silver compounds are being used to aid the process of wound healing, leading to the development of silver sulphadiazine exhibiting broad-spectrum antibacterial activity [69]. Silver is inert by nature and ionizes rapidly when it comes in contact with open environmental condition, forming Ag^+^ ions which are believed to produce antimicrobial activity [70]. Because of its strong antimicrobial activity, silver is a mostly used as an additional therapy in wound care. However, it also delays the process of infection and its spreading’s by producing toxic effects on the site of infection in keratinocytes and fibroblasts [71]. Nowadays, nanotechnology is a flourishing scientific field, focused on utilizing metal and metal oxide nanoparticles with extraordinary functions and size-dependent physicochemical properties, differing significantly from their macroscopic forms, in medicine. The continuous development of a new combination of nanomaterial, polymers, and antibiotics combined with search for methods for reduction of silver to obtain essential shapes of nanoparticles is promising way of revolutionizing treatments of wounds and AMR.

## 4. Nanotechnology and Wound Healing

Nanotechnology is a rapidly expanding discipline that combines material science and engineering. It deals with the understanding and control of matter at dimensions between 1-100nm. Nanoparticles have distinct physicochemical, optical, and biological properties that may be employed for a variety of applications. Metals, polymers, polysaccharides, and plant-derived bioactive compounds can be chemically formed into nanoparticles that can be combined with active drugs and used against human pathogens such as bacteria and viruses, as well as used to treat various anatomical and physiological conditions like cancer, hemophilia, stroke, blood disorder, and so on [72,73]. The utilization of nanotechnology materials is one of the most promising approaches for the invention of antibacterial agents and treatment processes [74]. 

Compared with the traditional wound healing, agents such as metal and metal oxide nanomaterials are more favorable because they harbor better intrinsic qualities, such as catalytic, optical, and melting properties [75]. The nano morphology, surface properties, porosity, and the ability of metals to resist decomposition in aqueous solutions, contribute to their efficacy in biomedical applications [76,77,78]. Many studies have explored the usefulness of nanoparticles and how they might be employed in medication delivery, diagnostics and imaging, biosensors, and cosmetics [79,80]. 

## 5. Mechanistic Insight into Silver Nanoparticles (AgNPs)

On average the concentration of silver in human plasma is less than 2 μg/mL, which is derived from inhalation of particulate matter and diet; dietary supplements, water contamination, or food fish and other aquatic organisms constitutes the potential sources of oral exposure [81]. Silver in ionic form can enter the human body through inhalation, oral ingestion and dermal absorption via wounds [67]. Two processes, pinocytosis and endocytosis are believed to be carry AgNPs inside the body. It has been noticed that the particles that are at nanoscale penetrate much deeper than those of bulk size, leading to develop a novel drug delivery system [82]. Till now, the exact mechanism of action of AgNPs is not yet clear, however several explanations of their antimicrobial properties have been proposed. Continual release of silver in its ionic form is considered to be the only reason for its antimicrobial activity [83]. Due to sulfur protein affinity and electrostatic attraction, silver ions adhere to the cells wall and cytoplasmic membrane, which increases its permeability, leading to the disruption and degeneration of the bacterial cell. When the silver ion enters the bacterial cell, it basically deactivates the respiratory enzymes and generate reactive oxygen species. The cell membrane disruption and DNA damage (by interacting with sulfur and phosphorus in the DNA molecule) causing problem in replication and reproduction, resulting in death of the microbes mainly due to reactive oxygen species which acts as a key component in the mechanism of action for silver [84].

Silver ions also inhibit the production of proteins by denaturing ribosomes and cause interruption in the production of ATP [83]. After anchoring and monitoring the surface of the cell, silver nanoparticles get accumulated in the pits of the cellular wall of microbes resulting in cell membrane denaturation and degeneration. Due to their nano size, they easily penetrate cell membrane, leading to rupture of cell organelles and even cell lysis. They also affect the bacterial transduction process by interfering with the phosphorylation of protein substrates which can result in cell apoptosis and multiplication. Gram-negative bacterial strains are more sensitive towards the effect of AgNPs because the cellular walls of these bacteria are narrower than those of Gram-positive bacteria [83]. 

One major drawback of silver nanoparticles is that they are not much effective and penetrating in case of bacterial biofilms. Biofilms generally protect the membrane from both silver ions and nanoparticles by altering the transport chain due to its complex structure. The ongoing pathway of the nanoparticle’s penetration is highly obstructed due to size of the nanoparticles i.e., ~50 nm. It has also been noticed that adsorption and accumulation of the silver nanoparticles on the bacterial biofilm results in reduced diffusion potential of nanoparticles in bacteria (Figure 3).

## 6. AgNPs Associated Wound Dressings

The active quest for alternate strategies of antibacterial defense has been driven by the decreasing effectiveness of present biocides and antibiotics, which has led to increasing mortality and higher medical expenses [85]. The use of metal nanoparticles, which eliminate bacteria by microbicidal and microbiostatic effects, is one of the most innovative methods [86,87]. Ag nanoparticles have proven to be the most successful of them all. For instance, silver nanoparticle-coated “cyborg” microorganisms are a very promising material that can be utilized to create smart antibacterial coatings in addition to delivering nanoparticles into organisms [85,88]. Collagen deposition, angiogenesis, epithelialization, granulation tissue development, and wound contraction all play important roles in the intricate process of wound healing. Silver nanoparticles (AgNPs) have been utilized extensively for wound healing in recent years [89]. Liu et al., used a full-thickness excisional wound model in mice for in vivo experimental methods in order to study the cellular response, epidermal re-epithelialization, and occurrence of dermal contraction during wound healing. They concluded that AgNPs could quicken the healing process [89,90]. In addition to these, silver nanocomposite films are also found to be very effective material for anti-bacterial application. Shevtsova et al., successfully fabricated the hybrid nanomaterials based on modified halloysite nanotubes (HNTs) containing grafted polymer brushes with silver nanoparticles. These temperature-responsive hybrid nanomaterials can be used to preserve solid substrates, make cutting-edge medical facemasks, and in the photothermal therapy against bacteria and tumors [91]. These composite nanoparticles demonstrated promising activity against pathogenic microbes at an in-vitro level and can cure wounds disinfections in-vivo. Despite having strong antibacterial properties, AgNPs exhibit cytotoxicity when exposed to cultured skin cells. AgNP-coated wound dressing’s clinical usefulness and safety are currently unknown. To assess the advantages of the antibacterial property against the hazard of toxicity when choosing medical techniques for wound therapy and management, professionals must have a better understanding of the toxic consequences of AgNPs during wound healing [92]. An overview of various dressing material fabricated with AgNPs available for wound dressing and their advantages are tabulated in Table 2.

### 6.1. AgNPs as Nanocomposite Material

The membrane and the diverse composite material-immobilized nanoparticles can have a plethora of different functions (e.g., disinfection, healing, catalytic activity). Silver has a bactericidal component that is commonly used to treat infection, burns, and different ulcers. Chronic non-healing lesions are treated with silver nitrate. AgNP-coated dressings, involved in the healing process, are now available in a variety of forms [93]. AgNPs are more hazardous at lower concentrations because their surface-to-volume ratio is higher. Pure silver nanoparticles can control the release of anti-inflammatory cytokines, allowing for faster wound healing without scarring [94]. By stimulating myofibroblast differentiation from normal fibroblasts, AgNPs aid to reduce the size of the infection and speed up the healing process. AgNPs also enhance epidermal re-epithelialization by stimulating keratinocyte proliferation and migration [95]. Szmyd et al., discovered that larger concentrations of AgNPs affect keratinocyte metabolism, migration, survival, and differentiation by activating caspases 3 and 7 (proteases implicated in programmed cell death) and causing dose-dependent DNA damage. AgNPs with tetracycline decreased bacterial load in the superficial and deep tissue layers of a mouse model, resulting in quicker wound healing [96]. For example, bio cellulose coupled with AgNPs worked as an antibacterial covering for open infections, with strong keratinocyte adhesion and proliferation around the wound margins. This nanomaterial has a high bacterial killing efficacy against Gram-negative pathogens and have shown to accelerate healing process [97]. 

Hanif et al. [98] described the in-situ reduction of silver salt layer by layer by using a non-toxic and environment friendly method involving tannic acid. The same methodology was adopted for the preparation of a cellulose-Ag nanocomposite with uniform and controlled size and distribution for the purpose of water disinfection, which was tested successfully for the presence of *E. coli* bacteria. In another study published by Dong et al. [99], casein-coated AgNPs were embedded onto acetate-cellulose membrane for the control of biofouling. The silver nanocomposite effectively suppressed the growth of *Serratia marcescens*, and it also profound the membrane activity and displayed a decrease in microbial growth by 59–99% after the lower concentration of silver nanoparticles were used [99]. Levi-Polyachenko et al. [100] by adopting solvent casting method, prepared chitosan membrane and loaded it with a different concentration of hexagonal AgNPs synthesized through the chemical approach to demonstrate the synergistic effect for wound healing activity [90]. The chitin membranes embedded with 100 ppm AgNPs showed promising antimicrobial activity against common pathogens (*P. aeruginosa* and *S. aureus*) [101]. 

### 6.2. AgNPs as Nanofibers

Nanofibers are the type of materials that can be used as a platform in raman spectroscopy [102], in immunoanalysis [103], for air filtration [104] and also as a pseudo-enzyme [105]. In a study, AgNPs coupled with poly(dopamine methacrylamide-co-methyl methacrylate) (MADO) nanofiber resulted in wound healing with the synthesis of epidermis on the wound area within two weeks, compared to AgNPs individually, which result in partial healing or required a long time to heal [106,107,108]. Y.C. Yeh and colleagues discovered that the Ag/AgCl/rGO nanomaterial improved healing with fast wound closure in mice burn lesions by improving epithelialisation and boosting collagen fiber deposition [108]. Lu et al. [109] also observed that inorganic particles like silica are persistently bound to open infections in animals. The novel chemical generated exhibits outstanding antibacterial characteristics while being less damaging to cells due to the di-sulphide connection between AgNPs and mesoporous silica nanoparticles. Very recently, El-Aassar et al. published a novel methodology to obtain silver-coated nanofibers by using polygalacturonic and hyaluronic acid. Successful AgNPs synthesis using PGA was verified timely by UV-vis spectral maxima ranging between 410 and 415 nm and through a transmission electron microscopic image confirming average particulate diameter of 8.6 nm. AgNPs components of (Ag-PGA/HA)-PVA nanofiber has produced robust zone inhibition confirming antibacterial activity against both Gram-positive and Gram-negative bacteria [110].

### 6.3. AgNPs as Hydrogels

Hydrogels have excellent ability to absorb exudates and at the same time also maintain the moisture content present in wound environment to ensure proper and rapid healing. Hydrogels prevent bacterial invasion from impermeable physical barrier on wound surface and showed its efficacy to absorb wide range of metals [105,111]. In a study reported that, AgNPs incorporated in poly(n-isopropylacrylamide-*co*-2-acrylamido-2-methylpropane sulfonic acid) (NIPAMSA) cryogel were used for the reduction of 4-nitrophenol to 4-aminophenol [112].

A new study found that higher levels of mercury exposure in young adults increased their risks for type 2 diabetes later in life by 65 percent. Therefore, AgNPs/starch/PEG/PAA hydrogel was created by Saberi et al., for Hg^2+^ removal. The maximum adsorption capacity of Hg^2+^ ions for hydrogel was found to be 158.21 mg/g and 182.53 mg/g in pH 7 and 6 in aqueous solutions, respectively [113]. Similarly, Dil and Sadeghi formulated nano-silver/gelatin/PAA hydrogel for Cu^2+^ removal. The absorption capacity was found to be 147.10 mg/g in pH 5.5 for 40 min when measured with the atomic absorption spectroscopy technique [114]. In a study, impregnated chitosan–PEG hydrogels were examined to assess and enhance the bio functionality of hydrogel materials for the delivery of AgNPs in diabetic patients suffering with chronic wounds [115].

### 6.4. AgNPs in Semi-Permeable Film Dressings

Semipermeable film dressings carry unique property of being flexible, thin, and transparent. The sheets of polyurethane or co-polyester covered with an adhesive layer allows the film dressing to adhere to the surface of skin [116,117]. Film dressings create a protective environment that is impermeable to microbes and liquids but permeable to water vapors, O_2_, and CO_2_. They can adhere to the place for one week. Semipermeable film dressings have been basically designed for superficial wounds, such as simple abrasions, minor burns, or lacerations [118,119]. New approaches and methodology regarding nanomaterial-based film dressings are being continuously investigated. Hubner et al. [120] recently concluded that novel gelatin-based films using glycerol act as a plasticizer when it gets incorporated with different concentrations of clinoptilolite zeolite impregnated with silver ions as wound dressings. Since these dressings are applied in both acute and chronic lesions treatment must carry antimicrobial characteristics. The silver-based compounds were used as antiseptics. For this purpose, films were produced by casting. All prepared concentrations of gelatin/clinoptilolite-Ag films confirmed the antibacterial activity against *S. aureus* and other human skin bacteria, not presenting meaningful differences in the size of the formed halo. Furthermore, Ambrogi et al. [121] also studied two different alginate films containing pyrogenic silica-supported with silver nanoparticles, which proved to have excellent hydration properties and a prolonged silver nanoparticles release, revealed it as potential wound dressings. The obtained films showed excellent antimicrobial and antibiofilm activities against *S. aureus* and *P. aeruginosa*. They displayed no such cytotoxicity towards human fibroblasts HuDe and human skin keratinocytes.

**Table 2 molecules-27-05587-t002:** An overview of available wound dressing materials fabricated with AgNPs and their advantage.

Wound Dressing Materials	Size of AgNPs (nm)	Microorganism	In Vivo/In Vitro Model	Advantage of Nanocoating	Reference
Polycaprolactone/Gelatin (PCLGelAg)	9–15	*S. aureus* and *P. aeruginosa*	Mice model	Membrane coated dressings revealed more significant antibacterial activities compared to single coating.	[92]
Chitosan/Poly(Ethylene Oxide) matrix	5	*E. coli*	-	Introduction of AgNPs enhanced the antibacterial activity based on their shape and size	[122]
Chitosan-Poly Vinyl Pyrrolidone (PVP) composite	10–30	*E. coli* and *S. aureus*	1929 cell line	Silver nanocomposite reduced the number of inflammatory cells by 99 in comparison to the control sample	[123]
Silver Alginate/Nicotinamide Nanocomposites	20–80	*E. coli* and *S. aureus*	Mice	Wound healing was achieved significantly after 4th day of treatment	[124]
Silver-Chitosan NPs-L-Glutamic Acid/Hyaluronic Acid	5–30	*E. coli* and *S. aureus*	Rabbit	Nanoparticle based natural matrix showed less inflammation in wounds compared with control after 15 days.	[125]
Cellulose hydrogel	5–50	*E. coli* and *S. aureus*	New Zealand rabbit	The average time for wound healing was 3 days in advance nanohydrogel compared to the control	[126]
Chitosan nanofiber	25	*S. aureus*	Wistar Hannover rats	The release of silver was significantly influenced by biological media: proteins created a barrier to silver release, whereas inorganic ions caused a sluggish release. As a result, inclusion of a large number of Ag-NPs was necessary to produce in vivo antibacterial effects.	[127]
Silver NPs embedded Bacterial cellulose gel membranes	30	*S. aureus*	Westar rats	After fourteen days of treatment, the wound healed (85.92%) significantly.	[128]
Chitosan-based multifunctional hydrogel	250	*E. coli* and *S. aureus*	Rat model	After 14 of treatment, the test organism exhibited lowest re-epithelialization rate	[129]
Chitosan-PEG hydrogel	75	*E. coli*, *P. aeruginosa* and *S. aureus*	Rabbit	At day fourteen, the Ag-NPs impregnated chitosan-PEG hydrogel group showed a healthy layer of dermal skin and a mixed pattern of collagen.	[115]
Chitosan cross-linked bilayer nanocomposite	45	*E. coli*, *P. aeruginosa* and *S. aureus*	L929 cell line	In comparison to the control group, the sustained-grown epithelium in the treatment group was more orderly and mature.	[130]
Asymmetric Wettable Chitosan nanocomposite	25	*E. coli*, *P. aeruginosa* and *S. aureus*	HEK293 cell line	An in vitro cytocompatibility investigation demonstrates that the dressing promotes cell development.	[131]
Polyvinyl-Pyrrolidone-Coated Silver Nanoparticles	10	*E. coli* and *S. aureus*	Mouse fibroblast (L929) cell line	Silver nanoparticles incorporated in PVP hydrogel led to cell enlargement.	[132]
Chitosan gels	15	*P. aeruginosa*	Human dermal fibroblasts	The evaluation of biocompatibility on primary fibroblasts revealed better results when the chitosan gels with Ag-NPs were analyzed	[133]
β-chitin-based hydrogels	5	*E. coli* and *S. aureus*	ERO cell line	The fabricated scaffolds displayed a greater capacity for whole blood clotting.	[134]
Hyaluronan Nanofiber	25	*E. coli* and *S. aureus*	Cell line (NIH 3T3)	Smaller particles have a greater impact on microorganisms, as evidenced by the nanoparticles size.	[135]
Activated Carbon coated silver nanocomposite	50–400	*S. aureus*, *Klebsiella pneumoniae* and *P. aeruginosa*	-	The Ag composites showed an increase in antibacterial activity when compared to the neat, activated carbon.	[136]
Polyurethane Foam mixed Ag-NPs Dressing	100	*E. coli*, *P. aeruginosa* and *S. aureus*	Human fibroblast	The foam dressing showed improved wound healing	[137]

## 7. Summary

Wounds are cured easily, but when infected, they can lead major damage to tissue and muscle. Such development of lesions leads to its chronic form which remains a major challenge as therapies that are unable to provide favorable outcome. Nowadays biological and synthetic nanomaterials are used to treat various types of inflammatory and pathogenic infection, and wound healing process. In this study we summarized the current scenario about AgNPs in wound care and medications. The purpose of this review study was to bring more attention to bactericidal mechanism of silver ions against variety of bacterial species. AgNPs have the capacity to destroy germs while also promoting skin regeneration. This unique property suggests that they can both effectively prevent wound infections and improve the healing process of damaged tissues in comparison with traditional topical treatments. Common wound dressing substances like chitosan, PVA, cellulose, and polyca-prolactone that have a variety of various topologies, such as nanofibers, hydrogels, and semi-permeable film, can be integrated with AgNPs.

The meticulous approach of silver nanoparticles and its versatility have provided the medical community with a new generation of highly effective and multi-dimensional treatment systems. It is essential to accomplish further preclinical studies to include the benefits of silver nanoparticles in tissue regeneration. Finally, fabrication of high purity nanoparticles is still a challenge as well, as often the bulk synthesis and purification of the nanoparticles and polymers are not facile. Thus, there is a continuous demand for improved synthetic and analytical techniques that will allow for the translation of nanotechnology-based methods to the clinic. However, attention should be paid to assess development of AMR towards any new product that is developed. Further research needs to be performed to identify new dressing materials conjugated with silver nanoparticles and their performances in accelerating chronic wound healing.

## Figures and Tables

**Figure 1 molecules-27-05587-f001:**
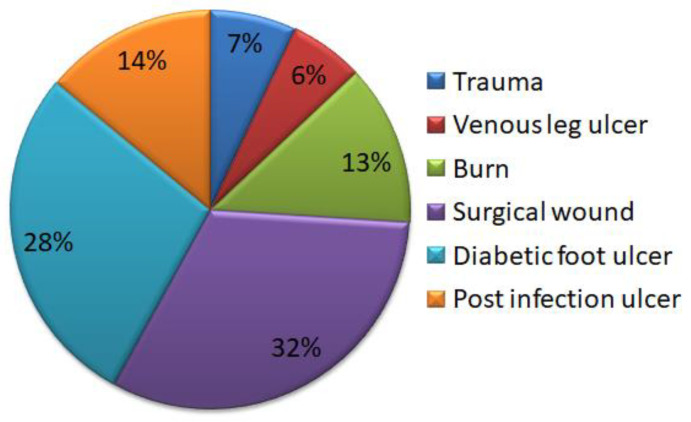
Pie chart representation on percentage of wound resources.

**Figure 2 molecules-27-05587-f002:**
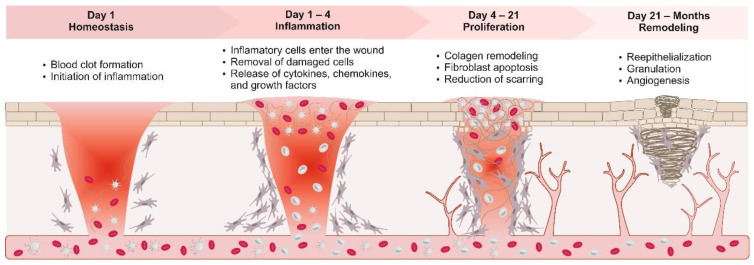
Visualization of the wound healing process and its outcomes.

**Figure 3 molecules-27-05587-f003:**
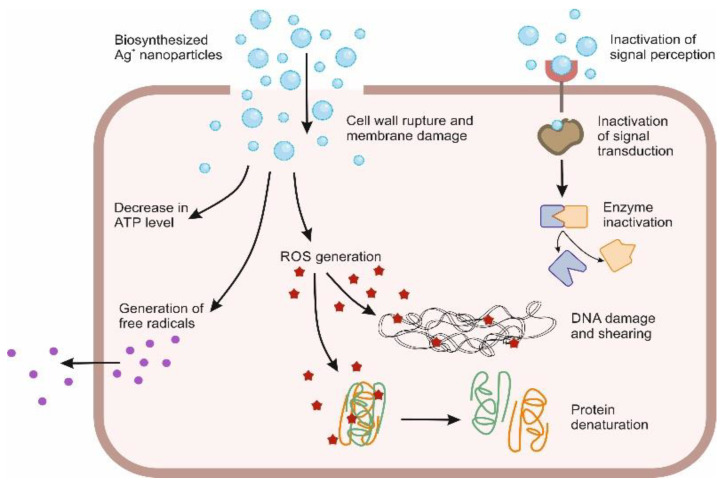
Biofunctionalized AgNPs as topical bullet for enhanced delivery and promising technology for wound healing.

**Table 1 molecules-27-05587-t001:** The description of some common wound infections and their traditional treatments.

Causative Bacteria	Wound Types	Diseases/Infections Associated	Preventive Measures	Drugs	References
*Staphylococcus* *aureus*	Acute	Abscesses (boils), Furuncles, Cellulitis	Maintaining good hygiene and regular and frequent hand washing	β-lactam antibiotics, Vancomycin, Daptomycin, Linezolid, Rifampin, and Tedizolid.	[25,26,27,28,29,30,31,32,33]
*Escherichia coli*	Clinical	Surgical site infections, Neonatal omphalitis and necrotizing fasciitis	Wash hands before handling, serving, or eating food, and especially after touching animals, working with livestock	Ciprofloxacin, Amoxicillin, Colistin, Tetracycline, Gentamicin and Cefuroxime	[19,34,35,36,37,38]
*Pseudomonas aeruginosa*	Open	Chronic wounds, pneumonia and UTIs	1% acetic acid is a simple, safe, and effective topical antiseptic that can be used in the elimination of *P. aeruginosa* from chronic infected wounds	Ciprofloxacin, Gentamicin and Kanamycin	[19,39,40,41]
*Klebsiellia pneumonia*	Chronic	UTIs	Strict adherence to hand hygiene, wearing gowns and gloves	Meropenem and Vaborbactam	[42]
*Streptococcus pyogens*	Acute	Strep throat, pharyngitis, scarlet fever (rash), impetigo, cellulitis, or erysipelas.	Wash hands before handling, serving, or eating food	Penicillin	[43]
*Proteus species*	Surgical acute	UTIs	Minimizing the incidence of infection using urinary catheterization and using high spectrum antibiotics	Ciprofloxacin	[44]
*Enterococcus faecalis*	Surgical	Bacteremia, UTIs, catheter-related infections, pelvic infections.	Practicing good hygiene and using potent antibiotics	Ampicillin Cefepime, Ceftaroline and Daptomycin	[45,46,47,48]

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
