# Peer review of "Silver Nanoparticles and Its Mechanistic Insight for Chronic Wound Healing: Review on Recent Progress"

_molecules, 2022, doi:10.3390/molecules27175587_

Round 1

Reviewer 1 Report (New Reviewer)

Comments

regarding the manuscript “Silver nanoparticles and its mechanistic insight for chronic wound healing: Review on recent progress” by Manoj Singh, Vanita Thakur, Vikas Kumar, Mayank Raj, Shivani Gupta, Nisha Devi, Sushil Kumar Upadhyay, Marketa Macho, Avik Baneerjee, Daniela Ewe and Kumar Saurav.

The aim of the presented review is to summarize the work to date on the use of silver nanoparticles in wound healing processes.

The paper consists of 9 pages of text + 5 pages of references. It should focus on the use of silver nanoparticles in wound healing, while its arrangement is significantly disturbed. The first five pages of the article describe the types of wounds, how to heal them, etc. which is far too detailed description. The main topic of the review article has been focused in the chapter 6, which is placed on the page 7. This is too large disproportion that does not allow the article to be rated highly. The subsections describing nanocomposites, nanofibers, and AgNPS hydrogels should be much more extended, they should contain tables listing various possibilities of using nanoparticles in wound healing. The topic was treated too vague and briefly and in my opinion the paper should not be published in this form.

Below I also present my technical comments regarding the article:

1)      In the Introduction section, the word "wounds" appears too often, the text should be changed so that it is not overused.

2)      There is also a lack of summary information on how silver nanoparticles can be obtained, which should be included in the text. This information does appear in particular sub-sections (e.g. line 256), but putting all the methods together would help the reader of the article

3)      The summary of the article should also be more detailed and actually summarize the collected literature.

Line 225, nancomposite should be changed to nanocomposite

Line 274 the polymer name: poly (Dopamine Methacrylamide-co-methyl methaacrylate) should be corrected

Line 296 the polymer name: poly (N-iospropylacryamide-co-acrylamido-2-methylpropane sulfonic acid) should be corrected.

Author Response

Regarding the manuscript “Silver nanoparticles and its mechanistic insight for chronic wound healing: Review on recent progress” by Manoj Singh, Vanita Thakur, Vikas Kumar, Mayank Raj, Shivani Gupta, Nisha Devi, Sushil Kumar Upadhyay, Marketa Macho, Avik Baneerjee, Daniela Ewe and Kumar Saurav. The aim of the presented review is to summarize the work to date on the use of silver nanoparticles in wound healing processes.

The paper consists of 9 pages of text + 5 pages of references. It should focus on the use of silver nanoparticles in wound healing, while its arrangement is significantly disturbed. The first five pages of the article describe the types of wounds, how to heal them, etc. which is far too detailed description. The main topic of the review article has been focused in the chapter 6, which is placed on the page 7. This is too large disproportion that does not allow the article to be rated highly. The subsections describing nanocomposites, nanofibers, and AgNPS hydrogels should be much more extended, they should contain tables listing various possibilities of using nanoparticles in wound healing. The topic was treated too vague and briefly and in my opinion the paper should not be published in this form.

Response: Authors thank the reviewer for the constructive response to improve the quality of the review article. We have made proper changes and hope that all concerns raised are addressed. With due respect, we would also like to bring into your attention that, we had been advised by editorial board to cut short our review and focus only on silver nanoparticle before the independent review process. But we understand your concern and tried our best to extend the subsections related to silver nanoparticle. We have also included a table summarizing an overview of available wound dressing materials fabricated with silver nanoparticles and their advantage.

Below I also present my technical comments regarding the article:

1. In the Introduction section, the word "wounds" appears too often, the text should be changed so that it is not overused.

Response: As suggested, we have avoided the repetition of the wound word.

2. There is also a lack of summary information on how silver nanoparticles can be obtained, which should be included in the text. This information does appear in particular sub-sections (e.g. line 256), but putting all the methods together would help the reader of the article.

Response: Authors thank the reviewer for their concern, and we would like to state that these methods were used as reference confirming the development of AgNPs as nanocomposite material. We have avoided using addition of section stating various methods used for AgNPs synthesis as these steps are very widely reviewed and presented elsewhere. We mainly tried to focus on AgNPs mechanistic insight for chronic wound healing. Under section 6, information has been updated for better understanding.

3. The summary of the article should also be more detailed and actually summarize the collected literature.

Response: Authors thank the reviewer for the suggestion, and we have modified the summary of the article based on the literatures reviewed.

4. Line 225, nancomposite should be changed to nanocomposite

Response: Changed as suggested.

5. Line 274 the polymer name: poly (Dopamine Methacrylamide-co-methyl methaacrylate) should be corrected

Response: Corrected as suggested.

6. Line 296 the polymer name: poly (N-iospropylacryamide-co-acrylamido-2-methylpropane sulfonic acid) should be corrected.

Response: Corrected as suggested.

Reviewer 2 Report (New Reviewer)

The Review "Silver nanoparticles and its mechanistic insight for chronic wound healing: Review on recent progress" summarized materials about the impact of silver nanoparticles on chronic wound healing. The review presents interesting results and can be published in Molecules mdpi after major revision.

The following essential moments should be clarified. 

1. Information about the toxicity of the silver nanoparticle and safe approaches for their application in would healing is absent in Review.

2. Please add information on the fabrication of the composite materials with silver nanoparticles for wound healing.

3. What about "smart" systems with silver nanoparticles as an advanced approach in medicine?

4. Please cite the following papers where relevant results were presented.

https://doi.org/10.1016/j.colsurfa.2022.128525

https://doi.org/10.1016/j.msec.2019.109806

https://doi.org/10.1016/j.jtemb.2021.126774

Author Response

The Review "Silver nanoparticles and its mechanistic insight for chronic wound healing: Review on recent progress" summarized materials about the impact of silver nanoparticles on chronic wound healing. The review presents interesting results and can be published in Molecules mdpi after major revision.

Response: Authors thank the reviewer for the constructive response to improve the quality of the review article. We have made proper changes and hope that all concerns raised are addressed.

The following essential moments should be clarified. 

1. Information about the toxicity of the silver nanoparticle and safe approaches for their application in would healing is absent in Review.

Response: Authors thank the reviewer for the suggestion, and we have included the relevant information on toxicity of AgNPs in the section 6.

2. Please add information on the fabrication of the composite materials with silver nanoparticles for wound healing.

Response: Authors thank the reviewer for the suggestion, and we have included the relevant information on the fabrication of the composite materials under section 6.

3. What about "smart" systems with silver nanoparticles as an advanced approach in medicine?

Response: Authors thank the reviewer for the suggestion, and we have included the information under section 6 with its relevant references.

4. Please cite the following papers where relevant results were presented.

https://doi.org/10.1016/j.colsurfa.2022.128525

https://doi.org/10.1016/j.msec.2019.109806

https://doi.org/10.1016/j.jtemb.2021.126774

Response: All the above-mentioned references are cited in the revised version of the manuscript at appropriate places.

Round 2

Reviewer 1 Report (New Reviewer)

The authors responded to all comments, therefore I believe that the article should be accepted for publication.

Reviewer 2 Report (New Reviewer)

The authors have addressed all my comments for this paper, I think the paper can be accepted for publication in its present form. 

This manuscript is a resubmission of an earlier submission. The following is a list of the peer review reports and author responses from that submission.

Round 1

Reviewer 1 Report

The manuscript entitled "Nanoparticles impregnated wound dressing material and its mechanical insight for chronic wound healing: Recent progress" is an overview of the applications of nanocomposite in wound healing, its mechanisms of action, as well as wound care scenarios. The review is written in clear language, logically and consistently. It is quite easy to read, while it is of scientific interest. The article can be accepted after minor changes.
1. Line 79. On Figure 1, the colors almost do not differ from each other, it is advisable to change the color saturation of the diagram.
2. Line 238. Table1 is separated from the description.
3. Lines 210, 237, 294, 313, 405, 423, 460, 470, 493, 557, 572, 593, 675, 692 - perhaps I should have deleted the lines.
4. Perhaps it would be necessary to divide the text into paragraphs, for easier perception. For example, Line 112.
5. I recommend that you add to Line 115 or Line 138 a link to a new article about silver nanoparticles, https://doi.org/10.3390/mi12121480

Author Response

File attached

Reviewer 2 Report

The manucript entitled " Nanoparticles impregnated wound dressing material and its mechanistic insight for chronic wound healing: Recent progress" is an interesting approach  at the dressings used wound healing. It is full of important and constructive information. Hovewer in some places is chaotic. Some information should be more precise, verified and completed. Manuscript shoul be better organized. Please find all comments in the text of manuscript.

Author Response

File attached
